# Identify Candidate Genes Associated with the Weight and Egg Quality Traits in Wenshui Green Shell-Laying Chickens by the Copy Number Variation-Based Genome-Wide Association Study

**DOI:** 10.3390/vetsci11020076

**Published:** 2024-02-06

**Authors:** Suozhou Yang, Chao Ning, Cheng Yang, Wenqiang Li, Qin Zhang, Dan Wang, Hui Tang

**Affiliations:** 1Key Laboratory of Efficient Utilization of Non-Grain Feed Resources (Co-Construction by Ministry and Province), Ministry of Agriculture and Rural Affairs, Shandong Agricultural University, 61 Daizong Street, Tai’an 271018, China; 2022120494@sdau.edu.cn (S.Y.); ningchao@sdau.edu.cn (C.N.); 2022010090@sdau.edu.cn (C.Y.); 2021010079@sdau.edu.cn (W.L.); 2Shandong Provincial Key Laboratory of Animal Biotechnology and Disease Control and Prevention, Shandong Agricultural University, 61 Daizong Street, Tai’an 271018, China; qzhang@sdau.edu.cn; 3College of Animal Science and Technology, China Agricultural University, Beijing 100083, China

**Keywords:** Wenshui green shell-laying chickens, weight and egg quality traits, copy number variation, genome-wide association study

## Abstract

**Simple Summary:**

The Wenshui green shell-laying chicken is an improved breed crossed from the Wenshang reed-feather chicken and Xinyang green shell-laying chicken, whose major characteristics are reed feathers, green-shelled eggs, a high egg-laying number, and excellent egg quality. The study of body weight and egg quality traits in Wenshui green shell-laying chickens is important for chicken-related breeding work. In this paper, we performed a copy number variation regions (CNVRs)-based genome-wide association study (GWAS) in Wenshui green shell-laying chickens to identify variations and candidate genes associated with their weight and egg quality traits. Finally, we identified important genes associated with body weight and egg quality traits. This study can provide a basic reference for the genetic improvement of chickens’ body weight and egg quality traits.

**Abstract:**

Copy number variation (CNV), as an essential source of genetic variation, can have an impact on gene expression, genetic diversity, disease susceptibility, and species evolution in animals. To better understand the weight and egg quality traits of chickens, this paper aimed to detect CNVs in Wenshui green shell-laying chickens and conduct a copy number variation regions (CNVRs)-based genome-wide association study (GWAS) to identify variants and candidate genes associated with their weight and egg quality traits to support related breeding efforts. In our paper, we identified 11,035 CNVRs in Wenshui green shell-laying chickens, which collectively spanned a length of 13.1 Mb, representing approximately 1.4% of its autosomal genome. Out of these CNVRs, there were 10,446 loss types, 491 gain types, and 98 mixed types. Notably, two CNVRs showed significant correlations with egg quality, while four CNVRs exhibited significant associations with body weight. These significant CNVRs are located on chromosome 4. Further analysis identified potential candidate genes that influence weight and egg quality traits, including *FAM184B*, *MED28*, *LAP3*, *ATOH8*, *ST3GAL5*, *LDB2*, and *SORCS2*. In this paper, the CNV map of the Wenshui green shell-laying chicken genome was constructed for the first time through population genotyping. Additionally, CNVRs can be employed as molecular markers to genetically improve chickens’ weight and egg quality traits.

## 1. Introduction

CNV is one of the common structural variation phenomena in the genome, ranging in size from 50 bp to several Mb. Its variation types include copy number deletions, insertions, recombinations, and multi-site complex mutations [1]. CNV is also one of the significant genetic bases for the evolution of individual phenotypic diversity and population adaptation [2]. It accounts for a relatively large proportion of the total genetic variation in a species [3], usually through dosage and positional effects of genes to achieve structural variation in genes [4,5]. It can modulate organismal plasticity and influence disease production and development [6]. It is widespread in the genomes of humans and other species, covers many more nucleotides than the total number of single nucleotide polymorphisms (SNPs), and greatly enriches the diversity of genomic variation [7].

In the current poultry production process, genetic variation has received widespread attention as one of the main factors influencing traits as generations alternate. Among them are several studies of CNV. For example, identified CNVs have been associated with broiler body weight [8], belly fat [9], and skin color [10], as well as breed-specific CNVs detected at the population level [11]. Therefore, studying chicken traits from a genomic perspective can help further develop their economic traits.

Currently, there are many methods to investigate CNV. In addition to conventional cytogenetic research methods, other methods include Array Comparative Genomic Hybridization (aCGH) chip technology [12], SNP chip technology [13], and next-generation sequencing (NGS) technology [14]. NGS technology, compared to aCGH chip technology and SNP chip technology, has higher resolution, the ability to perform diversity analysis, and the capability to detect a wider range of variations [15]. In NGS, various software can analyze CNV detection from whole-genome sequencing (WGS) data. According to the principle, software can be divided into four categories, namely Read-pair (RP), Split-read (SR), Read-depth (RD), and Assembly (AS); there is also some software using the Combined Approach (CA) to detect CNV [16,17].

With the popularization of CNV research in animals, many researchers have attempted to perform CNV-based GWAS research [2,8,18,19,20,21]. Since the concept of the GWAS was first proposed by Risch et al. in 1996 [22], the GWAS has been used primarily to discover genes associated with human genetic disease [23]. The genome information of many animals has improved with the rapid development of sequencing technology. GWAS research on CNV has gradually shifted from human diseases to economic and phenotypic traits in livestock species [18,19,24,25,26]. This indicates that CNV may significantly impact critical economic traits of livestock [27].

This paper aimed to identify CNV in Wenshui green shell-laying chickens and to conduct GWAS analysis for weight and egg quality traits based on copy number variation regions (CNVRs). The aim was to explore the genetic variation and candidate genes related to weight and egg quality traits of Wenshui green shell-laying chickens and to provide the basis for applying molecular breeding techniques such as molecular marker-assisted selection and genome selection to improve chickens’ weight and egg quality traits.

## 2. Materials and Methods

### 2.1. Population Description

The animals in this experiment were selected from Jinqiu Agricultural and Animal Husbandry Technology Co., Ltd. (Tai’an, China), and a total of 834 Wenshui green shell-laying chickens from the same batch were selected as the experimental group. Egg production data were recorded for 3 months after the test group started laying, and egg quality was measured for eggs laid at 30 and 40 weeks of age. The Wenshui green shell-laying chicken selected for this experiment is an improved breed. It is a cross between the Wenshang reed-feather chicken and the Xinyang green shell-laying chicken, created through selection technology that combines modern conventional breeding technology and molecular detection technology. The varieties are cultivated by aggregating the good characters (genes) such as reed feathers, green-shelled eggs, high egg yield number, and good egg quality.

### 2.2. Phenotyping

The egg weight and egg shape index of Wenshui green shell-laying chickens at 30 weeks of age were measured. The egg weight (EW), egg shape index (ESI), yolk color (YC), egg white height (EWH), shell thickness (SH), shell strength (SS), yolk weight (YW), shell weight (SW), egg white weight (EWW), yolk ratio (YR), shell ratio (SR), egg white ratio (EWR), concentrated egg white long diameter (EWL), concentrated egg white short diameter (EWS) and Haugh unit (HU) of Wenshui green shell-laying chickens at 40 weeks of age were measured. The weights of Wenshui green shell-laying chickens at birth (BW), 4 weeks (4-W), 8 weeks (8-W), 13 weeks (13-W), 15 weeks (15-W), and 38 weeks (38-W) of age were measured. In order to ensure the quality of the data, the phenotypic data were quality tested and the detailed results are shown in Appendix A.

Blood was collected from the test group at around 50 weeks of age using the subwing venous blood collection method. Insulation was prepared before collection, to prevent samples from going bad due to high temperatures. The storage temperature was −20 °C. TLANGEN’s Genomic DNA Extraction Kit was used to extract DNA from blood. Each sample’s DNA was extracted from the blood using the phenol–chloroform procedure, and the samples were analyzed for DNA contamination using a 1% agarose gel [28]. The OD_260_/OD_280_ ratio was measured to identify the content and quality of the DNA, with the OD_260_/OD_280_ generally ranging from 1.7–1.9. The extracted high-quality DNA was sent to Beijing Youji Technology Co., Ltd. (Beijing, China) for whole-genome sequencing (paired-end sequencing was performed and reads were 150 bp in length) to obtain the raw genomic data.

### 2.3. Sequence Alignment to Reference Genome

Quality screening of the raw genomic data was carried out by removing adapter sequences using Trimmomatic software v0.38 [29]. The average reading per sample after quality control was 44,931,409 and the average sequencing depth was 11.74X. The quality- screened data were compared to the reference genome (the reference genome is the Wenshui green shell-laying chicken’s genome) using bwa software v0.7.17 [30]. The average comparison rate was as high as 99.76%, and the average coverage was 97.55%. Repeat sequences were labeled using GATK software v4.2.6.1 after a comparison of the data [31].

### 2.4. CNV Detection

In this paper, DELLY software v1.1.6 was used in the CA method to detect CNV in the genomic data of the test population. DELLY combines short-range and long-range paired-end mapping and split-read analysis to detect balanced and unbalanced forms of structural variation, such as deletions, tandem duplications, inversions, and translocations, with high sensitivity and specificity [32]. Since DELLY software v1.1.6 is designed for detecting CNVs in populations, the population results directly present the curated population’s CNVR (the copy number variation region is a large genomic segment formed by adjacent copy number variation sections with overlapping regions [33]) obtained by merging overlapping CNVs across samples. The CNVR is defined as gain, loss, or mixed (losses and gains happening in the same area). The CNVs detected by DELLY were filtered by retaining the rows that contained “pass” in the FILTER column of the vcf file. Additionally, CNVRs within the range of 50 bp to 5 Mb were selected. After applying these filters, the CNVRs were further examined using BEDTools software v2.26.0. The purpose was to ensure that any overlapping CNVs in the population were merged [34].

### 2.5. CNV-Based GWAS

We selected the CNVR datasets with frequencies above 0.5% in each population to increase the precision of the GWAS results [18]. Format conversion of collated vcf files to bim, fam, and bed files was performed using plink v2.0 [35], where the bed files storing the genotypes were stored and needed to be recoded. Coding was based on genotype information in vcf files, gain, loss, and normal (2n), and coded as 1, −1, and 0, respectively [8,19]. The mixed linear model (MLM) of GMAT software was selected for single-trait genome-wide association analysis [36]. The individuals measured in this paper were born during the same period and housed in the same coop in stepped cages for rearing. The uniform model is as follows:y=μ+Wg+Zu+e
where y is the vector of phenotypic observations for individuals; μ is the population mean; g is the individual CNV effect vector and W is the design matrix of g; u is the multigene effect vector and Z is the design matrix of u; e is the random residual vector. Where the random effects follow a normal distribution,
$$u\sim N\left(0,G\sigma \frac{2}{a}\right),e\sim N(0,I\sigma \frac{2}{e})$$
where G is the genomic relationship matrix constructed by CNV; I is the unit array; $\sigma \frac{2}{a}$ is the additive genetic variance; $\sigma \frac{2}{e}$ is the random residuals. In the CNVR-based GWAS, we established a threshold for genome-wide significance, which was set at (0.05/N). N represents the number of CNVRs [18].

### 2.6. Gene Annotation

Self-written scripts were developed, considering the genes contained in a 100 kb window (50 kb up- and downstream) around the genomic regions of significant associated CNVRs based on the gff file of the Wenshui green shell-laying chicken’s reference genome.

## 3. Results

### 3.1. Number and Distribution of CNVR

CNV testing was carried out on the same batch of 834 Wenshui green shell-laying chickens using DELLY software v1.1.6. By combining overlapping CNVRs in all individuals, a total of 11,035 CNVRs were obtained with a total length of 13.1 Mb, representing approximately 1.4% of the chicken genome. There were 10,446 loss types, 491 gain types, and 98 mixed types (loss and gain within the same region). The CNVR sizes ranged from 51 bp to 642.6 kb, with an average CNVR length of 1.2 kb. Among the autosomes, the highest number of CNVRs was found on chromosome 1 (2761) and the lowest number on chromosome 35 (2). Among the chromosomes, chromosomes 29, 32, and 37 had no CNVRs. The reason for this may be the different lengths of each chromosome. The distribution of total CNVRs on various autosomes is shown in Table 1 and Figure 1 and Figure 2. Details about CNVRs are present in Appendix A.

In total, 9645 out of 110,35 CNVRs (87.4%) had sizes within the 0.05–5 kb interval, followed by 1060 (9.6%) within 1–5 kb, 146 (1.3%) within 5–10 kb, 160 (1.5%) within 10–50 kb, and 24 (0.2%) were greater than 50 kb in length (Figure 3).

### 3.2. CNVR-Based Genome-Wide Association Study

The genome-wide association study (GWAS) was performed to identify associations between the CNVR and 23 weight and egg quality traits evaluated in Wenshui green shell-laying chickens. Five traits were found to be significantly associated with the CNVR. A Manhattan plot of the CNVRs significantly associated with weight and egg quality traits on the 38 autosomes is shown in Figure 4. Table 2 lists the CNVRs significantly associated with weight and egg quality traits, among which two CNVRs were significantly correlated with 30-EW, one CNVR was significantly correlated with 40-EW and 40-EWW, four CNVRs were significantly correlated with 8-W, and one CNVR was significantly correlated with 38-W. Eleven genes were identified within a 100-kb window in genomic regions defined by significant CNVRs associated with 30-EW, 40-EW, 40-EWW, 8-W, and 38-W.

## 4. Discussion

In this paper, we identified 11,035 CNVRs in Wenshui green shell-laying chickens with a total length of 13.1 Mb, representing approximately 1.4% of their autosomal genome. There were 10,446 loss types, 491 gain types, and 98 mixed types. CNVR sizes ranged from 51 bp to 642.6 kb. To compare these results to previous studies, for example, Rao et al. detected 357 CNVRs using PennCNV software to detect CNVs in F2 generation flocks of White Recessive Rock and Xinhua chickens. Among the CNVRs, there were 213 loss types, 112 gain types, and 32 mixed types [37]. Chen et al. combined a variety of CNV detection methods; mrFAST, CNVnator, BreakDancer, and Pindel were tested against one original (Red Jungle fowl), two commercial (Recessive White Rock chickens, White Leghorn chickens) and local Chinese chickens (Xinghua chickens, Luxi Game fowl, Beijing-You chickens). A total of 11,123 CNVRs were detected, of which 8834 were loss types, 1911 gain types, and 378 mixed types [38]. Seol et al. used CNVnator software v0.4 to identify CNVs in four species (Cornish chickens, White Leghorn chickens, Rhode Island Red chickens, Red Jungle fowl) and screened 3079 CNVRs, of which 2443 were loss types and 636 were gain types [11]. Zhang et al. used PennCNV software to identify CNVs in the 475th generation of broiler lines from Northeast Agricultural University and screened 460 CNVRs, of which 320 were loss types, 93 gain types, and 47 mixed types [9]. Han et al. used aCGH microarray technology to detect CNVs in five breeds of chickens (Xichuan black-bone chickens, Silkie chickens, Lushi chickens, Gushi chickens, and Houdan chickens) and screened 281 CNVRs, among which 181 were loss types, 91 gain types, 9 mixed types [10]. When comparing these results, it was found that the CNVR replication rate detected in the studies was not high, which could be due to a number of reasons, such as different methods (aCGH array, SNP array, and next-generation sequencing (NGS)), different species, different algorithmic software, and quality screening of results. In this paper, the results showed that there were more loss-type CNVs than gain-type and mixed-type CNVs, which is basically consistent with the results of other studies mentioned above. The reasons for this result may be multifaceted, and there is no specific explanation yet. It is speculated that it may be due to the differences in the genetic structure of the species and the differences in the selective pressure during the breeding process. In addition, in this paper, CNVs were not detected on some chromosomes, which may be due to various reasons. These reasons include the quality of the genomic data, the detection methods used, and the structure of the chromosomes. Among these factors, we have already conducted quality checks on the genomic data, and the detection method used in this experiment is widely accepted. Therefore, when considering multiple factors, it is possible that the structure of the chromosomes themselves is the reason.

After conducting a CNVR-based GWAS on weight and egg quality traits, it was found that six CNVRs on chromosome 4 were significantly correlated with weight and egg quality traits. Among them, two CNVRs were significantly correlated with egg quality traits. The genes screened were *FAM184B*, *MED28*, *LAP3*, *ATOH8*, and *ST3GAL5*. Four CNVRs were significantly correlated with body weight traits, and the genes screened were *LOC112532307*, *LDB2*, *LOC107053295*, *LOC121110716*, *SORCS2*, and *LOC121110591*.

The *FAM184B* gene is a protein-coding gene that is widely expressed in a variety of tissues, including the brain and skin. In chickens, Zhang et al. found that the *FAM184B* gene on chromosome 4 was associated with the first spawning weight in Jinghai yellow chickens [39]. Jin et al. found that the *FAM184B* gene on chromosome 4 was closely related to body weight in Yancheng chickens [40]. In this paper, 30-EW and 40-EW were associated with the *FAM184B* gene. One of the factors affecting egg weight is the weight of hens at the start of laying after sexual maturity, and the weight at the start of laying directly determines the weight of hens at the peak of laying [41,42,43]. These results suggest that the *FAM184B* gene indirectly affects egg weight during the laying period by influencing the body weight of the Wenshui green shell-laying chickens.

The *MED28* gene is a protein-coding gene involved in cell proliferation and cycle regulation. In cattle, the *MED28* gene was found to be associated with body weight and intramuscular fat content [44,45,46]. In sheep, the *MED28* gene was found to be associated with prenatal and postnatal body weights [47,48]. In pigs, the *MED28* gene was found to be associated with muscle development [49]. There are no reports in the literature about the *MED28* gene in chickens. In this paper, 30-EW and 40-EW were correlated with the *MED28* gene. It is hypothesized that the *MED28* gene indirectly influences egg weight during the laying period by influencing the body weight of the Wenshui green shell-laying chickens.

The *LAP3* gene encodes an aminopeptidase that catalyzes N-terminal amino acid removal and is implicated in protein maturation and degradation [50]. Liu et al. found that the *LAP3* gene on chromosome 4 in Beijing-You chickens is associated with carcass weight and visceral weight [51]. In this paper, 30-EW and 40-EW were correlated with the *LAP3* gene. It is hypothesized that the *LAP3* gene indirectly influences egg weight during the laying period by influencing the body weight of the Wenshui green shell-laying chickens.

The *ATOH8* gene is a transcription factor with a bHLH domain that is involved in the development of the nervous system, kidney, pancreas, retina, and muscle [52]. Studies in chickens have shown the expression of the *ATOH8* gene during skeletal myogenesis in chickens [53,54]. In mice, it has been found that the *ATOH8* gene regulates muscle cell proliferation by modulating myopeptide signaling [53]. The energy metabolism status of muscles is a primary factor influencing the quality of eggs [55]. In this paper, a correlation was found between the 30-EW and the *ATOH8* gene. This suggests that the *ATOH8* gene indirectly affects the egg weight during the laying period by influencing the growth of skeletal muscles in Wenshui green shell-laying chickens.

The *ST3GAL5* gene is a protein-coding gene that plays a crucial role in glycosylation reactions and is involved in biological processes such as immune regulation and neural system development [56,57]. Research conducted in chickens has found that the *ST3GAL5* gene exhibited activity toward lactosylceramide. Furthermore, it was observed to be expressed at relatively higher levels in the small intestine, large intestine, and spleen of chickens [58]. The small intestine and large intestine are vital organs for maintaining the digestive, endocrine, metabolic, and immune functions in livestock [59]. In this paper, a correlation was found between the 30-week egg weight (30-EW) and the *ST3GAL5* gene. This suggests that the *ST3GAL5* gene indirectly affects the egg weight during the laying period in Wenshui green shell-laying chickens by influencing their digestive system.

The *LDB2* gene may bind to a number of transcription factors and is critical for brain development and blood vessel creation [60,61]. In a paper on chickens, Gu et al. discovered a correlation between the *LDB2* gene on chromosome 4 and body weight during weeks 7–12 in an F2 population of Silky fowl and White Plymouth Rock chickens [62]. Zhang et al. and Wang et al. also found a correlation between the *LDB2* gene on chromosome 4 and body weight in Gushi-Anka F2 chickens and Jinghai Yellow chicken hens [63,64]. Liu et al. found that the *LDB2* gene on chromosome 4 in Beijing-You chickens is associated with carcass weight and visceral weight [51]. Dou et al. discovered that the *LDB2* gene is an important candidate gene influencing the rapid growth of broiler chickens [65]. In this paper, a correlation was found between the 8-week weight (8-W) and the LDB2 gene. This suggests that the *LDB2* gene can influence the body weight of Wenshui green shell-laying chickens.

The *SORCS2* gene is a member of the Vps10p protein family, and Vps10p is associated with neurological disorders in mammals. In a paper on chickens, Li et al. explored the genetic mechanisms associated with aggressive behavior in chickens. They found that the *SORCS2* gene on chromosome 4 in a Chinese native breed, a dwarf yellow meat-type chicken, may play an important role in regulating dopamine pathways and neurotrophic factors involved in aggressive behavior [66]. Chen et al. found that the *SORCS2* gene is a candidate gene for aggressive behavior in Luxi Game fowl [38]. In this paper, a correlation was found between the 8-week weight (8-W) and the *SORCS2* gene. This suggests that the *SORCS2* gene can influence the body weight of Wenshui green shell-laying chickens.

At present, this is the first CNV identification in Wenshui green shell-laying chickens. CNVs were identified and compiled into CNVRs, while only autosomal chromosomes were considered. The association between weight and egg quality traits and CNVRs was determined by the GWAS. The interpretation of CNVR adjacent genes is helpful to better understand the weight and egg quality traits of Wenshui green shell-laying chickens. In addition, the limitations of this paper lie in the fact that there are many genetic variation factors that influence animal traits, such as SNPs, structural variation (SV), CNV, and DNA methylation. This study only focused on CNV detection in the Wenshui green shell-laying chicken breed, and further research is needed to investigate other aspects of genetic variation. This would contribute to a more comprehensive understanding of the Wenshui green shell-laying chicken breed.

## 5. Conclusions

CNVs were identified for the first time in a population of Wenshui green shell-laying chickens and merged into CNVRs. GWAS analysis based on CNVRs was conducted to investigate their association with weight and egg quality traits. A total of 11,035 CNVRs were identified in the entire population, accounting for approximately 1.4% of the chicken autosomal genome. The paper identified *FAM184B*, *MED28*, *LAP3*, *ATOH8*, and *ST3GAL5* as potential candidate genes influencing 30-EW and 40-EW, while *LDB2* and *SORCS2* were identified as potential candidate genes influencing 8-W. Therefore, this research reveals the potential impacts of CNVs on weight and egg quality traits in Wenshui green shell-laying chickens, providing new insights for future studies.

## Figures and Tables

**Figure 1 vetsci-11-00076-f001:**
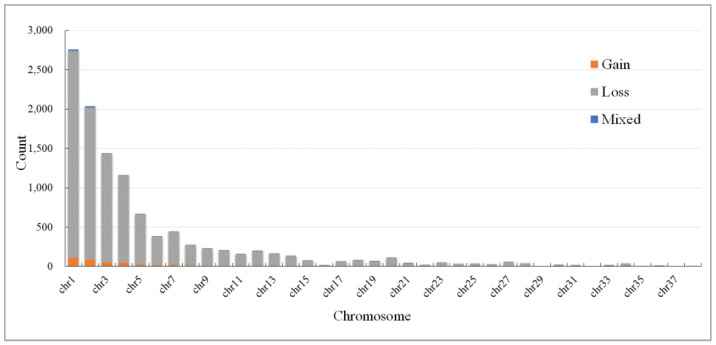
Distribution of CNVR types in Wenshui green shell-laying chickens.

**Figure 2 vetsci-11-00076-f002:**
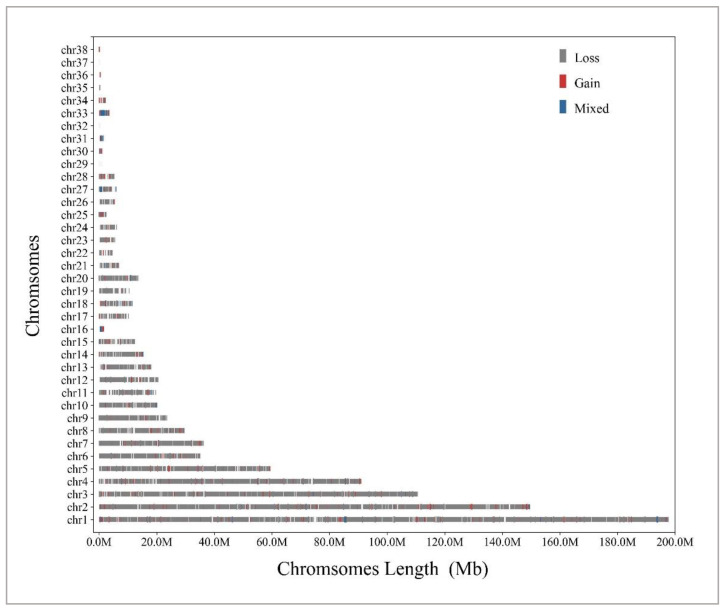
The overall CNVR maps for Wenshui green shell-laying chickens in the 38 autosomes. There are three distinct categories of CNVR: loss (grey), gain (red), and mixed (blue). Autosome values are on the Y-axis, and chromosomal location in Mb is on the X-axis.

**Figure 3 vetsci-11-00076-f003:**
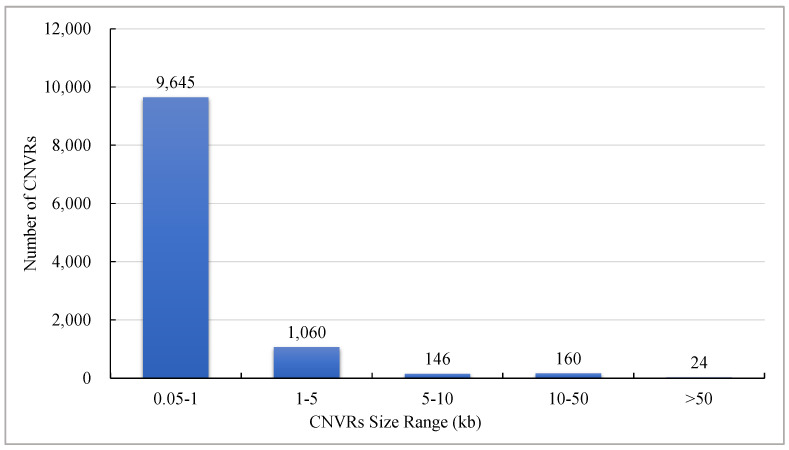
Distribution of CNVR sizes in Wenshui green shell-laying chickens.

**Figure 4 vetsci-11-00076-f004:**
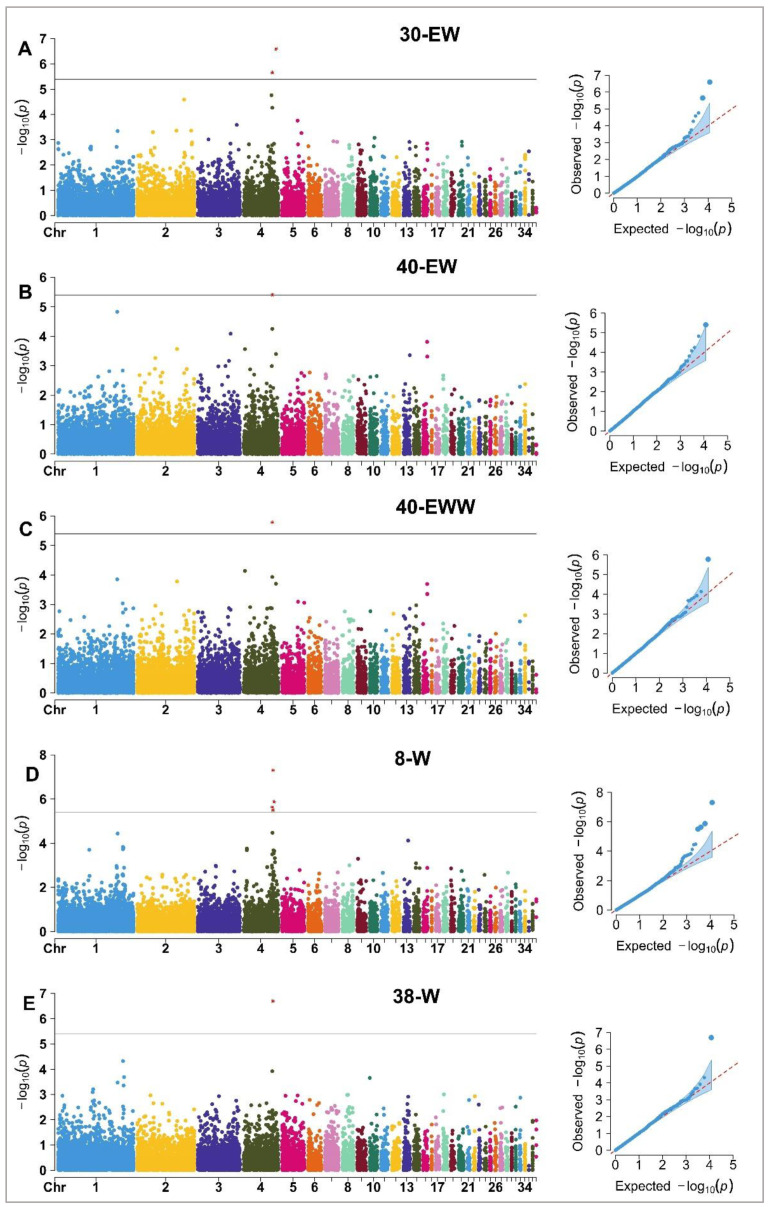
Manhattan plot and QQ plot for CNVR segments on the 38 autosomal chromosomes associated with 30-EW, 40-EW, 40-EWW, 8-W, and 38-W. In the Manhattan plot, the *X*-axis represents the autosome and the *Y*-axis indicates the −log10 (*p*-value). The lines in the chart indicate FDR-corrected *p*-values of 0.05. The Manhattan plot shows the CNVRs that are significantly as-sociated with the trait, while the QQ plot shows the significance of the association between CNVRs and the trait.

**Table 1 vetsci-11-00076-t001:** Distribution of CNVRs across autosomal chromosomes of the Wenshui green shell-laying chicken’s genome.

Chr	Chr Length (bp)	CNVR Count	Length of CNVR (bp)	Coverage (%)	Max Size (bp)	Average (bp)	Min Size (bp)
1	197,778,178	2761	4,102,658	2.1	642,753	1485.9	51
2	149,541,958	2038	1,795,942	1.2	76,049	881.2	51
3	110,815,227	1435	1,011,600	0.9	37,549	704.9	51
4	91,021,375	1162	894,849	1.0	104,295	770.1	51
5	59,471,259	664	581,860	1.0	38,456	876.3	51
6	35,339,061	382	241,170	0.7	21,044	631.3	51
7	36,318,844	437	353,483	1.0	36,899	808.9	51
8	29,613,760	276	143,465	0.5	20,456	519.8	52
9	23,556,363	224	177,489	0.8	15,124	792.4	51
10	20,214,400	204	122,553	0.6	12,825	600.8	51
11	19,755,808	157	81,894	0.4	16,494	521.6	51
12	20,438,972	194	124,811	0.6	19,162	643.4	51
13	18,437,548	161	285,249	1.5	50,758	1771.7	51
14	15,523,295	135	89,883	0.6	18,875	665.8	51
15	12,662,000	71	80,617	0.6	14,389	1135.5	51
16	1,595,800	11	615,070	38.5	542,075	55,915.5	63
17	10,229,956	57	22,414	0.2	4896	393.2	51
18	11,472,971	78	101,581	0.9	35,771	1302.3	51
19	10,411,340	66	65,454	0.6	12,765	991.7	51
20	14,040,156	110	78,193	0.6	16,021	710.8	51
21	6,776,000	42	36,126	0.5	27,274	860.1	52
22	4,690,381	23	15,414	0.3	4663	670.2	53
23	5,830,993	53	17,315	0.3	6364	326.7	52
24	6,352,200	37	16,508	0.3	6088	446.2	53
25	2,575,857	34	44,705	1.7	10,788	1314.9	52
26	5,288,600	30	23,279	0.4	13,044	776.0	51
27	5,930,361	56	425,849	7.2	311,709	7604.4	53
28	5,407,282	42	29,718	0.5	12,414	707.6	52
29	1,064,585	0	0	0	0	0	0
30	979,082	21	8930	0.9	1139	425.2	54
31	2,139,823	13	308,985	14.4	193,165	23,768.1	58
32	454,000	0	0	0	0	0	0
33	3,524,363	19	1,142,884	32.4	504,390	60,151.8	52
34	2,223,258	30	28,054	1.3	16,716	935.1	73
35	327,777	2	17,147	5.2	15,351	8573.5	1796
36	493,600	7	7529	1.5	3094	1075.6	226
37	316,000	0	0	0	0	0	0
38	298,400	3	1258	0.4	477	419.3	365
overall	6,247,088	11,035	13,093,936	1.4	642,753	1186.6	51

**Table 2 vetsci-11-00076-t002:** Significant CNVRs associated with weight and egg quality traits in Wenshui green shell-laying chickens.

Trait ^1^	CNVR ID	Type ^2^	Chromosome	CNVR Position (bp) ^3^	*p*-Value ^4^	Proximal Gene ^5^
30-EW	DUP00035918	Gain	4	85,203,669–85,205,350	2.52 × 10^−7^	ATOH8 ST3GAL5
30-EW 40-EW 40-EWW	DEL00035336	Loss	4	75,882,077–75,882,619	2.22 × 10^−6^ 3.93 × 10^−6^ 0.65 × 10^−6^	FAM184B MED28 LAP3
8-W	DEL00035339	Loss	4	76,019,681–76,020,779	2.35 × 10^−6^	LOC112532307 LDB2
8-W 38-W	DEL00035404	Loss	4	77,124,281–77,124,469	5.01 × 10^−8^ 2.02 × 10^−7^	LOC107053295
8-W	DEL00035425	Loss	4	77,492,948–77,493,042	3.16 × 10^−6^	LOC121110716
8-W	DEL00035578	Loss	4	80,144,379–80,145,908	1.35 × 10^−6^	SORCS2 LOC121110591

^1^ 30-EW: egg weight at 30 weeks of age; 40-EW: egg weight at 40 weeks of age; 40-EWW: egg white weight at 40 weeks of age; 8-W: weight at 8 weeks of age; 38-W: weight at 38 weeks of age. ^2^ Gain: duplications; loss: deletions. ^3^ CNVR Position: CNVR position based on the Wenshui green shell-laying chicken’s genome. ^4^ *p*-value: genome-wide significance. ^5^ Proximal Gene: ID of the gene on NCBI.

## Data Availability

The datasets presented in this study can be found in online repositories. The sequencing data used in the current study have been submitted to the NCBI Sequence Read Archive (SRA) under Bioproject: PRJNA1027008. However, the data are not yet public.

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
