# Peer review of "Identify Candidate Genes Associated with the Weight and Egg Quality Traits in Wenshui Green Shell-Laying Chickens by the Copy Number Variation-Based Genome-Wide Association Study"

_vetsci, 2024, doi:10.3390/vetsci11020076_

Round 1

Reviewer 1 Report

Comments and Suggestions for Authors

The paper entitled " Identify Candidate Genes Associated With the Weight and Egg Quality Traits in Wenshui Green-Shell Laying Chickens by the Copy Number Variation-Based Genome-Wide Association study " set out CNVs were identified for the first time in a population of Wenshui green shell laying chickens and merged into CNVRs. GWAS analysis based on CNVRs was conducted to investigate their association with weight and egg quality traits. The work is potentially interesting. The manuscript can be accepted after minor revisions.

 There comments for the manuscript are mentioned below:

1. In the introduction, the method of detecting CNV is too much described, but the status quo and significance of research on poultry CNV are lacking, and the research background is not clear and complete.

2.Why choose NGS technology over aCGH chip technology, SNP chip technology, and NGS technology?

3. Why did the subjects choose Wenshui green-shell laying chickens?

4. The clarity of Figure 2 needs to be increased.

5. "Fig3" or "Figure 3", the format must be the same.

6. The discussion needs improvement.

7. Please indicate the limitations of this study.

Comments on the Quality of English Language

English needs to be improved throughout, the manuscript would benefit from a thorough proofread by a native English speaker to improve.

Author Response

Response to the Comments of Submission vetsci-2797776

Journal

Veterinary Sciences

Manuscript ID

vetsci-2797776

Paper Title

Identify Candidate Genes Associated With the Weight and Egg Quality Traits in Wenshui Green-Shell Laying Chickens by the Copy Number Variation-Based Genome-Wide Association Study

Dear Editor in Chief, Editors and Reviewers,

First of all, we would like to express our thanks to the anonymous reviewers for their invaluable comments. We have found them helpful in improving the manuscript. Based on these comments, the manuscript has been thoroughly revised to address all issues raised in the reviews. In what follows, the details are provided.

Questions for General Evaluation

Reviewer’s Evaluation

Does the introduction provide sufficient background and include all relevant references?

Can be improve

Are all the cited references relevant to the research?

Yes

Is the research design appropriate?

Yes

Are the methods adequately described?

Yes

Are the results clearly presented?

Yes

Are the conclusions supported by the results?

Yes

 Point-by-point response to Comments and Suggestions for Authors

Comment 1: In the introduction, the method of detecting CNV is too much described, but the status quo and significance of research on poultry CNV are lacking, and the research background is not clear and complete.

Response: Done. We thank the reviewer’s valuable suggestion. According to the suggestion, We have modified the introduction by making corresponding deletions in the paragraphs describing the method of detecting CNV and adding an elaboration of the status quo and significance of research on poultry CNV to make the research background clear. The following are the details of the revision:

CNV is one of the common structural variation phenomena in the genome, ranging in size from 50 bp to several Mb. Its variation types include copy number deletions, insertions, recombinations, and multi-site complex mutations. CNV are also one of the significant genetic bases for the evolution of individual phenotypic diversity and population adaptation. It accounts for a relatively large proportion of the total genetic variation in a species, usually through dosage and positional effects of genes to achieve structural variation in genes. It can modulate organismal plasticity and influence disease pro-duction and developmen. It is widespread in the genomes of humans and other species, covers many more nucleotides than the total number of single nucleotide polymorphisms (SNP), and greatly enriches the diversity of genomic variation.

In the current poultry production process, genetic variation has received widespread attention as one of the main factors influencing traits as generations alternate. Among them are several studies of CNV. For example, identify CNVS associated with broiler body weight, belly fat, and skin color, as well as breed-specific CNVS detected at the population level. Therefore, studying chicken traits from a genomic perspective can help further develop their economic traits.

Currently, there are many methods to investigate CNV. In addition to conventional cytogenetic research methods, other methods include Array Comparative Genomic Hy-bridization (aCGH) chip technology, SNP chip technology and next-generation sequencing (NGS) technolog. NGS technology, compared to aCGH chip technology and SNP chip technology, has higher resolution, the ability to perform diversity analysis, and the capability to detect a wider range of variations. In NGS, various software can analyze CNV detection from whole genome sequencing (WGS) data. According to the principle of software can be divided into four categories, namely Read-pair (RP), Split-read (SR), Read-depth (RD) and Assembly (AS); there is also some software using the Combined approach (CA) to detect CNV.

With the popularisation of CNV research in animals, many researchers have at-tempted to perform CNV-based GWAS research. Since the concept of GWAS was first proposed by Risch et al. in 1996, GWAS has been used primarily to discover genes associated with human genetic disease. The genome information of many ani-mals has improved with the rapid development of sequencing technology. GWAS re-search on CNV has gradually shifted from human diseases to economic and phenotypic traits in livestock species. This indicates that CNV may significantly impact critical economic traits of livestock.

This paper aimed to identify CNV in Wenshui green-shell laying chickens and to conduct GWAS analysis for weight and egg quality traits based on CNVR. To explore the genetic variation and candidate genes related to weight and egg quality traits of Wenshui green-shell laying chickens and to provide the basis for applying molecular breeding techniques such as molecular marker-assisted selection and genome selection to improve chicken’s weight and egg quality traits.

Comment 2: Why choose NGS technology over aCGH chip technology, SNP chip technology, and NGS technology?

Response: Done. We deeply appreciate the reviewer for the valuable comment. NGS technology has the following advantages over aCGH microarray technology and SNP microarray technology in detecting CNVs:

  • High resolution: NGS technology can provide higher resolution because it can detect smaller CNVs.
  • Detection of a wider range of variants: NGS technology can detect a large number of single nucleotide variants, insertions, deletions and structural variants, as well as compound variants, whereas aCGH microarray and SNP microarray technology can only detect certain structural variants and compound variants.
  • Diversity analysis: NGS technology can be used for both demographic and individual analyses and can detect individual specificity and population diversity of CNVs
  • Scalability: NGS technology can analyse hundreds of samples in parallel and can be applied to a wide range of different types of biological samples.

Comment 3: Why did the subjects choose Wenshui green-shell laying chickens?

Response: Done. We deeply appreciate the reviewer for the valuable comment. A little explanation for the choice of experimental animal for this study was the Wenshui green-shell laying chickens:

The Wenshui green-shell laying chickens is a breed formed by crossbreeding the local breed of Wenshang Luhua Chickens in Shandong Province with the Xinyang green-shell laying chickens in co-operation between our laboratory and enterprises. This breed has inherited the excellent traits of the above two breeds, with reedy feathers, green eggs, high egg yield, excellent meat quality, good egg quality, etc. The study of these excellent traits provides an important theoretical basis for subsequent breeder selection.

Comment 4: The clarity of Figure 2 needs to be increased?

Response: Done. We deeply appreciate the reviewer for the valuable comment. Due to the large number of CNVRs detected, the results expressed in the figure cannot be clearly shown, if the picture is enlarged, it can be seen more clearly.

Comment 5: "Fig3" or "Figure 3", the format must be the same.

Response: Done. We deeply appreciate the reviewer for the valuable comment. we has been changed to be consistent in both places.

Comment 6: The discussion needs improvement.

Response: Done. We deeply appreciate the reviewer for the valuable comment. In the discussion section, we have added some explanations for the experimental results, which are as follows:

In this paper, the results showed that there were more loss type CNVs than gain type and mixed type CNVs, which is basically consistent with the results of other studies men-tioned above. The reasons for this result may be multifaceted, and there is no specific ex-planation yet. It is speculated that it may be due to the differences in the genetic structure of the species and the differences in the selective pressure during the breeding process. In addition, in this paper, CNVs were not detected on some chromosomes, which may be due to various reasons. These reasons include the quality of the genomic data, the detec-tion methods used, and the structure of the chromosomes. Among these factors, we have already conducted quality checks on the genomic data, and the detection method used in this experiment is widely accepted. Therefore, when considering multiple factors, it is possible that the structure of the chromosomes themselves is the reason.

Comment 7: Please indicate the limitations of this study.

Response: Done. We deeply appreciate the reviewer for the valuable comment. According to your suggestion, we have added a description of the limitations of the paper in the discussion section. Here's the translation of the specific content:

The limitations of this paper lie in the fact that there are many genetic variation factors that influence animal traits, such as SNP, Structural Variation (SV), CNV, and DNA methylation. This study only focused on CNV detection in the Wenshui green-shell laying chickens breed, and further research is needed to investigate other aspects of genetic variation. This would contribute to a more comprehensive understanding of the Wenshui green-shell laying chickens breed.

Reviewer 2 Report

Comments and Suggestions for Authors

A review of “Identify Candidate Genes Associated With the Weight and Egg  Quality Traits in Wenshui Green-Shell Laying Chickens by the  Copy Number Variation-Based Genome-Wide Association study” by Yang et al.

In this paper, the researchers performed a GWAS study in Wenshui green-shell laying chickens to determine genetic effects of CNV on egg quality traits and body weight. Please find below my detailed comments:

1)     Methods sections , subsection “Sequence Alignment to Reference Genome”- many details are missing here. How many reads were used per sample ? What was the coverage ? Later in the text a reference genome is mentioned but not detailed. Is there a reference genome for this species and did the authors use it ? which built (e.i. version) of the reference genome was used ?

2)     This study used 834 chickens for a GWAS. It is not clear if this sample size is large enough. Did the authors did any power calculation ? Please include this info in the text.

3)     Results, subsection 3.1  - it is mentioned that most of the CNV were in chromosome 1 , as chromosomes are numbered based on their size it is not surprising. It would be helpful to show number and or size of CNVs across the different chromosomes normalized to each chromosome size.

4)     Most of the CNVs detected were “loss” and not “gain” or “mix” . It looks somewhat surprising. Please discuss this phenomenon in the discussion section.

5)     Results :  “ This suggests that the SORCS2 gene influences the body weight of Wensui green-shell laying chickens by affecting their fighting behavior.”  Assuming that a CNV is affecting size based on the gene being associated with aggression and fighting behavior is far fetched assumption. It can be verse versa, because SORCS2 affects body size it allows individual chicken with this CNV to become more aggressive (as they tend to be bigger). I suggest removing this sentence or nuance it and suggesting alternative explanations.

6)     It is not clear if any method for multiple correction was used  and how. Did the authors a-priori selected a p-value (10^10) cutoff ? are p-value presented corrected for multiple comparisons?

7)     Minor: “Methods” > “phenotyping” at the end of the paragraph. I believe “resequencing should be “sequencing”.

8)     Minor: the abbreviation CNRV is mentioned in the abstract but explained only later in the Methods section, please explain CNRV in the 1st time it appears in the text for ease of read

Comments on the Quality of English Language

Quality of English seems adequate 

Author Response

Response to the Comments of Submission vetsci-2797776

Journal

Veterinary Sciences

Manuscript ID

vetsci-2797776

Paper Title

Identify Candidate Genes Associated With the Weight and Egg Quality Traits in Wenshui Green-Shell Laying Chickens by the Copy Number Variation-Based Genome-Wide Association Study

Dear Editor in Chief, Editors and Reviewers,

First of all, we would like to express our thanks to the anonymous reviewers for their invaluable comments. We have found them helpful in improving the manuscript. Based on these comments, the manuscript has been thoroughly revised to address all issues raised in the reviews. In what follows, the details are provided.

 Questions for General Evaluation

Reviewer’s Evaluation

Does the introduction provide sufficient background and include all relevant references?

Yes

Are all the cited references relevant to the research?

Yes

Is the research design appropriate?

Can be improved

Are the methods adequately described?

Must be improved

Are the results clearly presented?

Can be improved

Are the conclusions supported by the results?

Must be improved

Point-by-point response to Comments and Suggestions for Authors

Comment 1: Methods sections , subsection “Sequence Alignment to Reference Genome”- many details are missing here. How many reads were used per sample ? What was the coverage ? Later in the text a reference genome is mentioned but not detailed. Is there a reference genome for this species and did the authors use it ? which built (e.i. version) of the reference genome was used ?

Response: Done. We deeply appreciate the reviewer for the valuable comment. We have corrected the errors you have raised, here are the paragraph corrections:

Quality screening of the raw genomic data was carried out by removing adapter se-quences using Trimmomatic software v0.38. The average reading per sample after quality control was 44.93 Mb, and the average sequencing depth was 11.74X. The quality- screened data were compared to the reference genome(The reference genome is the Wenshui green-shell laying chickens genome) using bwa software v0.7.1. The aver-age comparison rate was as high as 99.76%, and the average coverage was 97.55%. Repeat sequences were labeled using GATK software v4.2.6.1 after a comparison of the data.

As the reference genome used was assembled in our laboratory, it is currently being uploaded to the database, and we will notify the editorial office as soon as the upload is successful. It is predicted that it can be cited before the publication of this paper. We apologise for any inconvenience this may cause.

Comment 2: This study used 834 chickens for a GWAS. It is not clear if this sample size is large enough. Did the authors did any power calculation ? Please include this info in the text.

Response: Done. We deeply appreciate the reviewer for the valuable comment. We did not calculate the sample size power because this sample size was taken from the company by direct sequencing. We were reading from a previous study on chickens, Zhang et al. which used 475 samples[1], the sample size of the study by Rao et al. was 554[2]. Forty-seven samples were used in the study by Chen et al[3]. Ten samples were used in the study by Han et al[4]. Compared to the above studies, the sample of 834 in this paper is sufficient to support the experiment. We will explain the above description in the paper.

  1. Zhang, H.; Du, Z.Q.; Dong, J.Q.; Wang, H.X.; Shi, H.Y.; Wang, N.; Wang, S.Z.; Li, H. Detection of genome-wide copy number variations in two chicken lines divergently selected for abdominal fat content. Bmc Genomics 2014, 15, doi:Artn 51710.1186/1471-2164-15-517.
  2. Rao, Y.S.; Li, J.; Zhang, R.; Lin, X.R.; Xu, J.G.; Xie, L.; Xu, Z.Q.; Wang, L.; Gan, J.K.; Xie, X.J.; et al. Copy number variation identification and analysis of the chicken genome using a 60K SNP BeadChip. Poult Sci 2016, 95, 1750-1756, doi:10.3382/ps/pew136.
  3. Chen, X.; Bai, X.; Liu, H.G.; Zhao, B.B.; Yan, Z.X.; Hou, Y.L.; Chu, Q. Population Genomic Sequencing Delineates Global Landscape of Copy Number Variations that Drive Domestication and Breed Formation of in Chicken. Front Genet 2022, 13, doi:ARTN 83039310.3389/fgene.2022.830393.
  4. Han, R.L.; Yang, P.K.; Tian, Y.D.; Wang, D.D.; Zhang, Z.X.; Wang, L.L.; Li, Z.J.; Jiang, R.R.; Kang, X.T. Identification and functional characterization of copy number variations in diverse chicken breeds. Bmc Genomics 2014, 15, doi:Artn 93410.1186/1471-2164-15-934.

Comment 3: Results, subsection 3.1 - it is mentioned that most of the CNV were in chromosome 1, as chromosomes are numbered based on their size it is not surprising. It would be helpful to show number and or size of CNVs across the different chromosomes normalized to each chromosome size.

Response: Done. We deeply appreciate the reviewer for the valuable comment. The text is only a general statement, and if we put it all on, we feel that some of it is not streamlined enough. Detailed information is in Supporting Information Table S1. If you think it is not suitable, we will modify it again.

Comment 4: Most of the CNVs detected were “loss” and not “gain” or “mix”. It looks somewhat surprising. Please discuss this phenomenon in the discussion section.

Response: Done. We deeply appreciate the reviewer for the valuable comment. Based on the suggestions you made, we added the following statement to the discussion:

In this paper, the results showed that there were more loss type CNVs than gain type and mixed type CNVs, which is generally consistent with other studies cited in the article. The reasons for this result may be multifaceted, and there is no specific explanation yet. It is speculated that it may be due to the differences in the genetic structure of the species and the differences in the selective pressure during the breeding process.

Comment 5: Results: “This suggests that the SORCS2 gene influences the body weight of Wensui green-shell laying chickens by affecting their fighting behavior.”  Assuming that a CNV is affecting size based on the gene being associated with aggression and fighting behavior is far fetched assumption. It can be verse versa, because SORCS2 affects body size it allows individual chicken with this CNV to become more aggressive (as they tend to be bigger). I suggest removing this sentence or nuance it and suggesting alternative explanations.

Response: Done. We deeply appreciate the reviewer for the valuable comment. We have made the following modifications to your suggestions:

This suggests that the SORCS2 gene affects the fighting behavior of Wensui green-shell laying chickens, leading to increased physical activity and consequential effects on body weight.

Comment 6: It is not clear if any method for multiple correction was used  and how. Did the authors a-priori selected a p-value (10^10) cutoff ? are p-value presented corrected for multiple comparisons?

Response: Done. We deeply appreciate the reviewer for the valuable comment. In the gwas analyses, we focus on the P-value threshold settings. No other multiple corrections were made. In the CNVR-based GWAS, establish the threshold for genome-wide significance which was set at (0.05/N). Where N represents the number of CNVRs.

Comment 7: Minor: “Methods” > “phenotyping” at the end of the paragraph. I believe “resequencing should be “sequencing”.

Response: Done. We deeply appreciate the reviewer for the valuable comment. We have amended the text by replacing "resequencing" with "sequencing".

Comment 8: Minor: the abbreviation CNVR is mentioned in the abstract but explained only later in the Methods section, please explain CNVR in the 1st time it appears in the text for ease of read.

Response: Done. We deeply appreciate the reviewer for the valuable comment. We have added an explanation of CNVR in the article.

Reviewer 3 Report

Comments and Suggestions for Authors

Dear Authors:

this is our review of the great paper:

Summary: This study had two main objectives: first, identify CNVs markers in Wenshui green-shell laying chickens; second, estimate associations between the CNVs and weight and egg quality traits by the means of GWAS. The strengths of this study are: is the first (or one of the firsts) studies of this type with Wenshui green-shell laying chickens. There are few studies that used CNVs and GWAS, specially in chicken.

General concept comments:

Article: The article has been presented in a very good format, well written, with very good references. The methods were adequate to respond their objectives. There were just a handful of typographical errors (which I mentioned in the comments). The major weakness of the paper is that the results raises some questions that were not responded or commented in the discussion section (all of them also written in my comments).

Comments by section:

Simple summary:

  • Line 17: please, remove the expression “so on”. It is too vague. Suggestion: “which major characteristics are reed feathers, green shell eggs, high eggs laying number and excellent egg quality”.

Abstract:

  • Line 25-26: the sentence “Copy number variation (CNV) as an essential source of genetic variation is associated with animal biological traits.” suggests that CNVs are only present in or are only associated with animal biological traits, which is not true. Please rewrite the sentence.

Introduction:

  • I miss references about CNV in chicken in the introduction section. You have many of them in the references and discussion sections but did not mentioned them in the introduction.

Materials and Methods

  • Did you make any sort of quality control of the samples for the GWAS?
  • I believe it’s important to mention that you removed sex chromosomes and briefly explain why.

Results:

  • Line 181: please correct “In tota” to “In total”.
  • Line 193: please, start the sentence with the word “Five” instead with the number five “5”.
  • Table 2: please, review the footnotes indexing/superscript (missing number 1 in the footnotes and the number 4 in the table is not superscripted).

Discussion:

  • Why more loss type than gain or mixed types?
  • Why more CNVs on the first chromosomes? Although easy to know why, I think it can help the reader, specially if he/she is a student, to mention/remind that the first chromosomes are longer than the rest and probably that’s why there are more CNVs in those ones compared to the others. Besides the length, is there another explanation?
  • Why some chromosomes do not have CNVs? Is it due to chance, chromosome short size or another reason?
  • Table 1: Min size of CNV is never less than 51. Why? Something related to the WGS method?
  • Table 1: Min CNV size on chromosome 35 is 1796, so much higher than in the other chromosomes. Why?
  • Why does only the chromosome 4 has significant CNVs?

References:

  • Reference 2 and 20 are the same. Please review your references.

Keep the good work!

Author Response

Response to the Comments of Submission vetsci-2797776

Journal

Veterinary Sciences

Manuscript ID

vetsci-2797776

Paper Title

Identify Candidate Genes Associated With the Weight and Egg Quality Traits in Wenshui Green-Shell Laying Chickens by the Copy Number Variation-Based Genome-Wide Association Study

Dear Editor in Chief, Editors and Reviewers,

First of all, we would like to express our thanks to the anonymous reviewers for their invaluable comments. We have found them helpful in improving the manuscript. Based on these comments, the manuscript has been thoroughly revised to address all issues raised in the reviews. In what follows, the details are provided.

Questions for General Evaluation

Reviewer’s Evaluation

Does the introduction provide sufficient background and include all relevant references?

Can be improved

Are all the cited references relevant to the research?

Yes

Is the research design appropriate?

Yes

Are the methods adequately described?

Yes

Are the results clearly presented?

Yes

Are the conclusions supported by the results?

Can be improve

Point-by-point response to Comments and Suggestions for Authors

Summary: This study had two main objectives: first, identify CNVs markers in Wenshui green-shell laying chickens; second, estimate associations between the CNVs and weight and egg quality traits by the means of GWAS. The strengths of this study are: is the first (or one of the firsts) studies of this type with Wenshui green-shell laying chickens. There are few studies that used CNVs and GWAS, specially in chicken.

General concept comments: Article: The article has been presented in a very good format, well written, with very good references. The methods were adequate to respond their objectives. There were just a handful of typographical errors (which I mentioned in the comments). The major weakness of the paper is that the results raises some questions that were not responded or commented in the discussion section (all of them also written in my comments).

Response: Done. We deeply appreciate the reviewer for the valuable comment. In response to your suggestions, we will make changes in the text. Below we will answer your comments in detail.

Comments by section:

Simple summary: Line 17: please, remove the expression “so on”. It is too vague. Suggestion: “which major characteristics are reed feathers, green shell eggs, high eggs laying number and excellent egg quality”.

Response: Done. We deeply appreciate the reviewer for the valuable comment. We have made changes according to your comments.

Abstract:

Comment 1: Line 25-26: the sentence “Copy number variation (CNV) as an essential source of genetic variation is associated with animal biological traits.” suggests that CNVs are only present in or are only associated with animal biological traits, which is not true. Please rewrite the sentence.

Response: Done. We deeply appreciate the reviewer for the valuable comment. We have made the following changes in response to your comments:

Copy number variation (CNV) as an essential source of genetic variation can have an impact on gene expression, genetic diversity, disease susceptibility and species evolution in animals.

Introduction:

Comment 1: I miss references about CNV in chicken in the introduction section. You have many of them in the references and discussion sections but did not mentioned them in the introduction.

Response: Done. We deeply appreciate the reviewer for the valuable comment. In response to your comments, we have included the following narrative in the introduction:

In the current poultry production process, genetic variation has received widespread attention as one of the main factors influencing traits as generations alternate. Among them are several studies of CNV. For example, identify CNVS associated with broiler body weight, belly fat, and skin color, as well as breed-specific CNVS detected at the population level. Therefore, studying chicken traits from a genomic perspective can help further develop their economic traits.

Materials and Methods:

Comment 1: Did you make any sort of quality control of the samples for the GWAS? I believe it’s important to mention that you removed sex chromosomes and briefly explain why.

Response: Done. We deeply appreciate the reviewer for the valuable comment. The following are specific responses:

First, we have performed quality control on the animal phenotypic data and CNV results required for the GWAS methodology. We have calculated the maximum, minimum, mean, standard deviation, and standard error of the phenotypic data to conduct overall screening of the measured phenotype. We filtered the detected CNVs by retaining rows in the FILTER column of the VCF file that contained "pass". Additionally, we selected CNVRs within the range of 50bp to 5mb and calculated the population frequency of these CNVRs.

Secondly, regarding the article showing only autosomes, we checked the CNV research on related poultry, and some of the key CNVs appeared on autosomes and not on sex chromosomes. And it was the same phenomenon when we performed the test. Therefore, we did not show the CNVs on the sex chromosomes in the results, and if you feel that this is not appropriate, we will add this part of the results.

Results:

Comment 1: Line 181: please correct “In tota” to “In total”. Line 193: please, start the sentence with the word “Five” instead with the number five “5”. Table 2: please, review the footnotes indexing/superscript (missing number 1 in the footnotes and the number 4 in the table is not superscripted).

Response: Done. We deeply appreciate the reviewer for the valuable comment. We have made changes according to your comments.

Discussion:

Comment 1: Why more loss type than gain or mixed types?

Response: Done. We deeply appreciate the reviewer for the valuable comment. In this paper, the results showed that there were more loss type CNVs than gain type and mixed type CNVs, which is generally consistent with other studies cited in the article. The reasons for this result may be multifaceted, and there is no specific explanation yet. It is speculated that it may be due to the differences in the genetic structure of the species and the differences in the selective pressure during the breeding process.

Comment 2: Why more CNVs on the first chromosomes? Although easy to know why, I think it can help the reader, specially if he/she is a student, to mention/remind that the first chromosomes are longer than the rest and probably that’s why there are more CNVs in those ones compared to the others. Besides the length, is there another explanation?

Response: Done. We deeply appreciate the reviewer for the valuable comment. We accept your suggestion to add a note to the first autosomal chromosome that has more CNVs on it than on the other autosomes: "probably because the first chromosome is longer than the others". Other than the length reason, it is not possible to provide a more valuable reason for the time being, as there is no reliable basis for it.

Comment 3: Why some chromosomes do not have CNVs? Is it due to chance, chromosome short size or another reason?

Response: Done. We deeply appreciate the reviewer for the valuable comment. There can be multiple reasons why CNVs are not detected on certain chromosomes. These reasons include the quality of genomic data, the detection method used, and the structure of the chromosomes. Among these factors, we have already conducted quality checks on the genomic data, and the detection method used in this experiment is widely accepted. Therefore, when considering multiple factors, it is possible that the structure of the chromosomes themselves is the reason.

Comment 4: Table 1: Min size of CNV is never less than 51. Why? Something related to the WGS method?

Response: Done. We deeply appreciate the reviewer for the valuable comment. The minimum size for CNV is set at 51bp because, during the quality control of detected CNV results, we have established a CNV size range from 50bp to 5Mb. The detection of both smaller and larger CNVs is currently based on immature technologies, which can lead to errors. Filtering out these smaller and larger CNVs can therefore improve the accuracy of the results.

Comment 5: Min CNV size on chromosome 35 is 1796, so much higher than in the other chromosomes. Why?

Response: Done. We deeply appreciate the reviewer for the valuable comment. The significant variation in the minimum values of CNVs across different chromosomes may be attributed to the distinct genetic characteristics of each chromosome, such as chromosome size, gene density, and repetitive sequences. The differences in gene distribution and genetic features on chromosomes can potentially lead to variations in the size and distribution of CNVs.

Comment 6: Why does only the chromosome 4 has significant CNVs?

Response: Done. We deeply appreciate the reviewer for the valuable comment. "We currently have no conclusive evidence to provide a specific explanation for the significant CNV found only on chromosome 4. We speculate that this may be due to the important impact of certain key genes or regulatory regions on specific traits on this chromosome. Further research and functional analysis will be needed to determine the mechanisms by which these CNVs affect specific traits.

References:

Comment 1: Reference 2 and 20 are the same. Please review your references.

Response: Done. We deeply appreciate the reviewer for the valuable comment. We have corrected the error.

Round 2

Reviewer 2 Report

Comments and Suggestions for Authors

A review of the revised version of “Identify Candidate Genes Associated With the Weight and Egg  Quality Traits in Wenshui Green-Shell Laying Chickens by the  Copy Number Variation-Based Genome-Wide Association study” by Yang et al.

The researchers addressed most of my concerns. I still have some comments that must be addressed in my opinion before this paper can be accepted for publication:

1)     In their response to comment #1: The authors mentioned that they had to do a de novo assembly of the Wenshui green-shell laying chickens genome. They said they hope to deposit this assembly in the future. Also, the sequencing data is presumably deposited at NCBI Bioprogect PRJNA1027008, but it is unavailable. Without having the raw sequencing data alongside this new genome assembly it is hard to say anything about the quality control of the work. This data should be available before publication.

2)     It is not clear how long were the reads and where they single- or pair- end reads. Please provide this info.

3)     In their response to comment #3: Supplemental Table 1 does not provide the analysis I was expecting. To claim there is an enrichment of CNVs etc. in a certain chromosome, as it is implied in the manuscript, the authors should do an observed/expected analysis were length of the chromosomes are taken into consideration (ie, normalize CNV frequency to chromosome length) to support their claim.

4)     In their response to comment #5: the authors revised to: “This suggests that the SORCS2 gene affects the fighting behavior of Wensui green-shell laying chickens, leading to increased physical activity and consequential effects on body weight.” This revision does not address my question and concern and does not discuss the two possibilities where growth affects aggressiveness or verse-versa.

Comments on the Quality of English Language

English if OK, I haven't  detected any major errors.

Author Response

Response to the Comments of Submission vetsci-2797776

Journal

Veterinary Sciences

Manuscript ID

vetsci-2797776

Paper Title

Identify Candidate Genes Associated With the Weight and Egg Quality Traits in Wenshui Green-Shell Laying Chickens by the Copy Number Variation-Based Genome-Wide Association Study

Dear Editor in Chief, Editors and Reviewers,

First of all, we would like to express our thanks to the anonymous reviewers for their invaluable comments. We have found them helpful in improving the manuscript. Based on these comments, the manuscript has been thoroughly revised to address all issues raised in the reviews. In what follows, the details are provided.

 Questions for General Evaluation

Reviewer’s Evaluation

Does the introduction provide sufficient background and include all relevant references?

Yes

Are all the cited references relevant to the research?

Yes

Is the research design appropriate?

Can be improved

Are the methods adequately described?

Can be improved

Are the results clearly presented?

Can be improved

Are the conclusions supported by the results?

Can be improved

Point-by-point response to Comments and Suggestions for Authors

Comment 1: In their response to comment #1: The authors mentioned that they had to do a de novo assembly of the Wenshui green-shell laying chickens genome. They said they hope to deposit this assembly in the future. Also, the sequencing data is presumably deposited at NCBI Bioprogect PRJNA1027008, but it is unavailable. Without having the raw sequencing data alongside this new genome assembly it is hard to say anything about the quality control of the work. This data should be available before publication.

Response: Done. We deeply appreciate the reviewer for the valuable comment. We will release the data before the paper is published.

Comment 2: It is not clear how long were the reads and where they single- or pair- end reads. Please provide this info.

Response: Done. We deeply appreciate the reviewer for the valuable comment. As per your suggestion, we have explained it in the text. We performed paired-end sequencing and the length of the reads was 150bp.

Comment 3: In their response to comment #3: Supplemental Table 1 does not provide the analysis I was expecting. To claim there is an enrichment of CNVs etc. in a certain chromosome, as it is implied in the manuscript, the authors should do an observed/expected analysis were length of the chromosomes are taken into consideration (ie, normalize CNV frequency to chromosome length) to support their claim.

Response: Done. We deeply appreciate the reviewer for the valuable comment. We believe that Table 1 combined with Figure1-2 shows that the chromosome length makes a difference in the number of CNVRs on each chromosome. This can be done without normalize. We will add in the text to explain the above reason. If you think it is not appropriate, we will listen to your suggestion and add it.

Comment 4: In their response to comment #5: the authors revised to: “This suggests that the SORCS2 gene affects the fighting behavior of Wensui green-shell laying chickens, leading to increased physical activity and consequential effects on body weight.” This revision does not address my question and concern and does not discuss the two possibilities where growth affects aggressiveness or verse-versa.

Response: Done. We deeply appreciate the reviewer for the valuable comment. "The SORCS2 gene affects the body weight of Wensui green-shell laying chickens by influencing their fighting behavior." Such a conclusion is, indeed, inappropriate. We have modified the text to state only that the SORCS2 gene is a potential candidate gene for influencing the body weight of Wensui green-shell laying chickens.
